# Association between Dental Treatment, Quality of Life, and Activity Limitation According to Masticatory Discomfort: Evidence from the Korean National Health and Nutrition Examination Survey (2013–2015)

**DOI:** 10.3390/ijerph17020547

**Published:** 2020-01-15

**Authors:** Hyun-Kyung Kang, Yu-Rin Kim

**Affiliations:** Department of Dental Hygiene, Silla University, 140 Baegyang-daero 700 Beon-gil, Sasang-gu, Busan 46958, Korea; icando@silla.ac.kr

**Keywords:** health, mastication, oral health, quality of life

## Abstract

People with masticatory discomfort are unable to consume a balanced diet, which impacts their general health. We studied the relationship between quality of life and dental care associated with masticatory discomfort. Data from Korea’s representative 6th Korea National Health and Nutrition Examination Survey (KNHANES) were used. Complex sampling analysis with the stratification variable, clustering variable, and weight was applied. Demographic and dental treatment characteristics and activity limitations were compared through chi-square tests. The comparison of quality of life according to masticatory discomfort was performed using linear regression. The risk of masticatory discomfort was high in people who did not undergo regular oral examinations and preventive and definitive caries treatment and in those who received periodontal, surgical, endodontic, or prosthetic treatments. Generally, people with masticatory discomfort engaged in less activity owing to other disorders like arthritis, rheumatism, and back, neck, and oral disease. People with masticatory discomfort scored low on quality of life. People who received regular oral examinations and preventive care had a low level of masticatory discomfort, and the treated persons had high masticatory discomfort. Therefore, in order to reduce masticatory discomfort, more diverse and active care should be provided for prevention, specifically regular oral examinations.

## 1. Introduction

Oral health is an important contributor to general health as it is significantly associated with nutritional intake [1]. To maintain good oral health, regular oral check-ups and preventive treatments are needed, and oral diseases must be treated early to prevent the development of more serious oral health issues.

Mastication refers to the process of grinding food into a smaller size and swallowing after mixing it with saliva. The subjective method of evaluating the masticatory ability is evaluation of the patients’ perception of their masticatory status, as “good” or “bad,” and evaluating food intake using the Food intake Ability (FIA) questionnaire [2]. The objective method is measuring the occlusal force distribution or strength under static conditions [3]. Thus, mastication and oral health are closely associated. Such masticatory ability can be assessed objectively and subjectively. The percentage of individuals experiencing masticatory difficulties due to oral health issues is as high as 20%–46% [4], and this statistic highlights the importance of the masticatory ability among the various functions of teeth. Individuals with masticatory discomfort are unable to eat a well-balanced meal [5] and present with an especially low intake of fruits and vegetables. Therefore, maintaining good oral health is vital for a well-balanced nutritional intake [6]. An increase in the number of missing teeth results in lower intake levels of all nutrients including proteins, phosphorus, iron, thiamine, and niacin [7]. Further, the intake level of iron, potassium, thiamine, riboflavin, niacin, and vitamin C is lower in edentulous individuals [8]. Therefore, masticatory discomfort could lead to malnutrition [9].

Various studies have reported that masticatory ability is associated with general health, mental health, and quality of life (QoL) because better masticatory ability results in higher body weight, vestibular function, and bone content [10]. Studies have also reported that masticatory impairment or poor oral health status could cause a decline in social efficacy due to limitations in indulging in social activities [11], and that individuals with more functional teeth and a higher masticatory ability showed higher Geriatric Oral Health Assessment Index (GOHAI) scores [12]. This indicates that masticatory ability plays a major social and psychological role [13]. Meanwhile, other studies have reported that masticatory ability affects brain function and identified its association with various cognitive functions including learning, recall, retention, and attention [14], which necessitates the need to find measures to improve masticatory ability and reduce masticatory discomfort. Therefore, to reduce masticatory discomfort, it is necessary to identify the level of masticatory discomfort not only according to the oral health status, which is an objective indicator, but also by the type of recent dental treatment. In addition, it is necessary to identify the effects on activity limitation and QoL according to subjective masticatory discomfort. Until now, most of the studies on mastication included the elderly population [4,5,11,12]. However, it is necessary to include all age groups with various oral health conditions rather than limiting it to the elderly population.

Therefore, the present study used the 6th Korea National Health and Nutrition Examination Survey (KNHANES) data, representing the Korean citizens. The study aimed to identify the dental treatments leading to masticatory discomfort and the effects of masticatory discomfort on activity limitation and QoL.

## 2. Materials and Methods

### 2.1. Study Subjects

The present study used data from the 6th edition of KNHANES, which is an annual survey conducted by the Korea Center for Disease Control and Prevention (KCDC). The data used consisted of “No” (0 points) or “Yes” (1 point) responses to questions about masticatory discomfort in a health questionnaire survey. The study used data from 13,511 survey participants; 4510 from 2013, 4428 from 2014, and 4573 from 2015. As per government-designated statistics (approval number: 117002) based on Article 17 of the Statistics Act, the first and second years of the 6th KNHANES received approval from the Institutional Review Board (IRB) of KCDC (2013-07CON-03-4C and 2013-12EXP-03-5C, respectively). However, surveys in the third and subsequent years were conducted without IRB review and approval since they were classified as research conducted directly by the government for public welfare, in accordance with Article 2.1 of the Bioethics and Safety Act and Article 2.2.1 of the Enforcement Decree of the Bioethics and Safety Act.

### 2.2. Demographic Characteristics

From the health questionnaire survey of KNHANES, data on sex, age, marital status, household income, and economic activity were included. Age groups consisted of 12–19, 20–39, 40–59, and ≥60 years, while household income was divided into quintiles of low, lower-middle, middle, upper-middle, and high. Economic activity is any individual activity related to the production and consumption of goods or services or the distribution of income or wealth. Therefore, the difference between doing an economic activity and not doing it is simply the difference between having an income.

### 2.3. Types of Dental Treatment

The oral health survey was reviewed to examine the association between masticatory discomfort and recent dental treatments. With regard to oral examination, oral disease prevention, caries treatment, periodontal treatment, dental pulp treatment, oral surgical treatment, and prosthetic treatment, 0 was assigned to “not received recent treatment” and 1 to “recently treated”.

### 2.4. Activity Limitation

The masticatory system has a complex structure that includes the temporomandibular joint, neuromuscular system, teeth, and supporting tissues, which mutually maintain a close physiological relationship. Moreover, they are susceptible to functional decline with age, and masticatory discomfort is highly associated with bone, muscle, and oral diseases associated with aging [15]. As for whether masticatory discomfort resulted in limiting the daily and social activities, 23 questions excluding “others” were investigated. Three questions were found to be related to “arthritis, rheumatism”, “back and throat problems”, and “dental and oral diseases” where 0 points was assigned to “No” and 1 point was assigned to “Yes”.

### 2.5. QoL (EQ-5D)

EQ-5D [16] could be applied to various health levels including clinical conditions and is a tool used to measure simple and overall health status. It was investigated with approval from the EuroQol Group (www.euroqol.org) and viewed as the best tool for measuring the Quality Adjusted Life Years. The tool consists of five categories: athletic ability, self-care, daily activities, pain/discomfort, and anxiety/depression. Lower scores indicate better health status.

### 2.6. Statistical Analysis

Data were analyzed using IBM SPSS ver. 25.0 (IBM Co., Armonk, NY, USA) and complex sampling analysis with stratification, clustering, and weighting was applied to all data analyses. Moreover, complex sample chi-square tests were used for comparisons of demographic characteristics, type of dental treatment, and activity limitation according to masticatory discomfort. Linear regression analysis adjusted for sex, age, marital status, household income, and economic activity was performed to investigate the impact of masticatory discomfort on QoL. In the analysis, “unknown”, “not applicable” and missing values were excluded. The number of subjects in all tables was given as unweighted frequency and the significance level of statistical testing was set to 0.05.

## 3. Results

### 3.1. Demographic Characteristics According to Masticatory Discomfort

The results of demographic characteristics according to masticatory discomfort showed that masticatory discomfort was higher among those who were single and those who do not participate in economic activities (*p* < 0.001, Table 1).

### 3.2. Dental Treatments According to Masticatory Discomfort

The risk of masticatory discomfort was lower in the oral prevention group than in the non-prevention group with an odds ratio (OR) of 0.517; lower in the caries treatment group than in the non-treatment group with an OR of 0.844; higher in the periodontal treatment group than in the non-treatment group with an OR of 2.074; higher in the dental pulp treatment group than in the non-treatment group with an OR of 1.736; higher in the oral surgical treatment group than in the non-treatment group with an OR of 2.093; and higher in the prosthetic treatment group than in the non-treatment group with an OR of 2.090 (*p* < 0.001, Table 2).

### 3.3. Comparison of Activity Limitation According to Masticatory Discomfort

The risk of masticatory discomfort was higher in the group with arthritis/rheumatism, and their activity limitation was higher than that in the group with no such problem, with an OR of 1.712; higher in the group with back and throat problems than in the group with no such problem, with an OR of 1.633; and higher in the group with dental and oral diseases than in the group with no such problem, with an OR of 5.274 (*p* < 0.01, Table 3).

### 3.4. Impact of Masticatory Discomfort on QoL

Analysis adjusted for sex, age, marital status, household income, and economic activity was performed to investigate the impact of masticatory discomfort on QoL. Impact on QoL associated with athletic ability, self-care, daily activities, pain/discomfort, and anxiety/depression was lower by 0.215, 0.081, 0.172, 0.239, and 0.144 times, respectively, in those without masticatory discomfort when compared to those with discomfort. Significant differences were found in all variables (*p* < 0.001, Table 4).

## 4. Discussion

It has been reported that the incidence of depression was 1.4 times higher in elderly individuals with a decline in masticatory function as it causes a decline in fitness due to the consequential limitation in the intake of meals in terms of quantity and quality, with a negative impact on articulation and esthetics [17]. It has also been reported that oral health indices associated with mastication have an impact on QoL [18]. Accordingly, the present study aimed to identify the types of dental treatment associated with masticatory discomfort and investigate the effects of masticatory discomfort on activity limitation and QoL.

According to the National Health and Nutrition Examination Survey (NHANES) by the Korea Institute for Health and Social Welfare, almost half (46.6%) of the elderly population reported masticatory discomfort [19]. However, in the present study, there were more people without than those with masticatory discomfort, regardless of the age. This is thought to have improved the oral health of the elderly individuals as implant and denture treatments were covered under insurance for those aged 65 years and above in Korea. Moreover, due to the nature of elderly individuals with their various systemic diseases, the results may also reflect a difference in relative perception, with greater importance placed on systemic diseases than masticatory discomfort. Thus, caution is needed when interpreting the results. Moreover, the frequency of masticatory discomfort was higher in those who were single and those who did not participate in economic activities, whereas the frequency of masticatory discomfort was lowest in the highest income groups. It is thought that masticatory discomfort is greater because it is difficult to receive dental care in the absence of a stable source of income. Such findings were consistent with those of another study that reported that healthcare-seeking behavior increased with an increase in income level and involvement in economic activities [20].

Masticatory discomfort develops in those with poor oral health status, and to maintain good oral health status, it is necessary to receive regular preventive and curative caries treatments. In the present study, the risk of masticatory discomfort was lower by 0.517-fold in the group that received preventive treatment than in the group that did not and by 0.844-fold in the group that received curative caries treatment. The risk of masticatory discomfort was lower in people who received treatment at pre-pathogenic, early pathogenic, and early disease stages, which belonged to primary and secondary prevention according to the principles of prevention of oral diseases. This shows us the importance of undertaking a comprehensive management of oral diseases from the ethical, economical, and public health aspects whenever possible, prioritizing secondary prevention over tertiary prevention and primary prevention over secondary prevention.

In contrast, the risk of masticatory discomfort was higher in people who received treatment at advanced disease and recovery stages, which was then classified as tertiary prevention. Specifically, the risk of masticatory discomfort was higher by 2.074-, 1.736-, 2.093-, and 2.090-folds in people who received periodontal, pulpal, surgical, and prosthetic treatment, respectively, than those who did not. It is believed that the reason for such results is the fact that people with masticatory discomfort have advanced oral disease; therefore, it was difficult for them to recover their masticatory function despite various dental treatments [21].

Masticatory discomfort is closely associated with social health since it could have a negative impact on esthetics and articulation due to missing teeth and lack of nutritional intake. Activity limitation, which is an important indicator of social health, refers to partial or complete limitation of daily activities due to morbidity [22]. The risk of masticatory discomfort was higher by 1.172- and 1.633-fold in groups with activity limitation caused by arthritis/rheumatism and back and throat problems than in the groups without such problems. Above all, the risk of masticatory discomfort was much higher (more than 5-fold) in the group with activity limitation caused by dental and oral disease than in the groups without such problems. The findings were consistent with various studies that reported that activity limitation is closely associated with oral diseases [23] since activity limitation presents many barriers to social activities, which could lead to decrease in social interaction. Other studies have also reported that a decrease in the number of natural teeth causes activity limitation and decline in QoL [24], while differences in QoL could be found with the use of dentures [25]. In addition, another study reported that QoL was higher in those who were able to chew rice, kimchee, beef, and peanut better [26]. Therefore, normal masticatory function is known to have a positive impact on QoL. The results of the present study agree with those of a previous study in which the impact on QoL was lower in the absence of masticatory discomfort. These results have already been found in several studies. However, most of the studies were conducted on elderly Koreans, and none of them covered all age groups. Therefore, our research on South Korea’s entire age group is significant. The limitations of the present study are as follows: only the health examination questionnaire survey data were used, and thus, there is a limitation to the generalizability of the findings. Moreover, since masticatory discomfort was assessed from a subjective perspective, an in-depth analysis using objective data is deemed necessary. Therefore, future studies should combine the results from oral examination results and health examination questionnaire survey results to examine the association of masticatory discomfort with oral health, nutritional, and systemic health statuses with a broader scope.

## 5. Conclusions

Based on the findings of the present study, the discomfort of masticatory tended to be reduced due to preventive and early dental treatment and management. When the severity of oral disease increased, the patient developed high masticatory discomfort. Moreover, masticatory discomfort was shown to be causing limitations in social activities and decline in QoL. Therefore, in order to reduce masticatory discomfort, more diverse and active care should be provided with regular oral examinations and preventive treatments that are a part of primary prevention.

## Figures and Tables

**Table 1 ijerph-17-00547-t001:** General characteristics of study subjects. Unit: *N* (%).

Characteristics		No Discomfort	Discomfort	*p*
Year	2013	3415 (76.5)	1095 (23.5)	0.361
	2014	3332 (74.6)	1096 (25.4)	
	2015	3456 (74.9)	1117 (25.1)	
Sex	Male	4389 (75.3)	1456 (24.7)	0.947
	Female	5814 (75.4)	1852 (24.6)	
	Total	10,203 (75.3)	3308 (24.7)	
Age	12–19	2340 (75.3)	767 (24.7)	0.728
	20–39	2040 (74.7)	703 (25.3)	
	40–59	2961 (76.0)	906 (24.0)	
	≥60	2862 (75.3)	932 (24.7)	
	Total	10,203 (75.3)	3308 (24.7)	
Marital status	Single	8282 (72.5)	3107 (27.5)	0.001
	Married	1920 (90.8)	20 1(9.2)	
	Total	10,202 (75.3)	3308 (24.7)	
Household income	Lower	2073 (75.3)	682 (24.7)	0.375
	Lower-middle	2108 (76.0)	647 (24.0)	
	Middle	1980 (74.4)	696 (25.6)	
	High-middle	1998 (74.1)	652 (25.9)	
	High	1974 (76.6)	617 (23.4)	
	Total	10,133 (75.3)	3294 (24.7)	
Economic activity	None	3859 (70.8)	1562 (29.2)	0.001
	Active	5963 (79.2)	1578 (20.8)	
	Total	9822 (75.7)	3140 (24.3)	

By complex sample chi-square test.

**Table 2 ijerph-17-00547-t002:** Comparison of dental treatment according to mastication discomfort.

Characteristics		No Discomfort	Discomfort	OR (95% CI)	*p*
Regular oral examination	Yes	3530 (72.5)	1319 (27.5)	0.967 (0.837–1.117)	0.647
	No	1342 (71.8)	508 (28.2)	1.00	
Oral prevention	Yes	2197 (79.7)	544 (20.3)	0.517 (0.454–0.589)	0.001
	No	2675 (67.0)	1283 (33.0)	1.00	
Caries treatment	Yes	1479 (74.7)	477 (25.3)	0.844 (0.732–0.973)	0.020
	No	3393 (71.3)	1350 (28.7)	1.00	
Periodontal treatment	Yes	1039 (60.7)	646 (39.3)	2.074 (1.813–2.373)	0.001
	No	3833 (76.2)	1181 (23.8)	1.00	
Dental pulp treatment	Yes	927 (63.2)	525 (36.8)	1.736 (1.499–2.011)	0.001
	No	3945 (74.9)	1302 (25.1)	1.00	
Oral surgical treatment	Yes	595 (58.5)	403 (41.5)	2.093 (1.775–2.467)	0.001
	No	4277 (74.7)	1424 (25.3)	1.00	
Prosthetic treatment	Yes	1034 (60.6)	635 (39.4)	2.090 (1.833–2.383)	0.001
	No	3838 (76.3)	1192 (23.7)	1.00	

By complex sample chi-square test (*N* = unweighted), OR: odds ratio; CI: 95% Confidence interval; *p*: *p* of OR.

**Table 3 ijerph-17-00547-t003:** Comparison of activity limitation due to masticatory discomfort.

Characteristics		No Discomfort	Discomfort	OR (95% CI)	*p*
Arthritis, rheumatism	Yes	89 (37.0)	142 (63.0)	1.712 (1.214–2.414)	0.002
	No	447 (50.1)	423 (49.9)	1.00	
Back and throat problems	Yes	128 (38.9)	183 (61.1)	1.633 (1.193–2.236)	0.002
	No	408 (51.0)	382 (49.0)	1.00	
Dental and oral diseases	Yes	4 (15.0)	21 (85.0)	5.274 (1.520–18.295)	0.004
	No	532 (48.2)	544 (51.8)	1.00	

By complex sample chi-square test (*N* = unweighted), OR: odds ratio; CI: 95% Confidence interval; *p*: *p* of OR.

**Table 4 ijerph-17-00547-t004:** Impact of masticatory discomfort on quality of life (QoL)

Characteristics			No Discomfort	
	Estimate (95% CI)	SE	T	*p*
Athletic ability ^a^	0.215 (0.237–0.193)	0.011	−19.018	<0.001
Self care ^b^	0.081 (0.096–0.067)	0.0.07	−11.034	<0.001
Daily activities ^c^	0.172 (0.195–0.150)	0.011	−15.265	<0.001
Pain/discomfort ^d^	0.239 (0.267–0.212)	0.014	−17.107	<0.001
Anxiety/depression ^e^	0.144 (0.165–0.123)	0.011	−13.355	<0.001

By complex sample linear regression analysis, discomfort = 1, R^2^ (*P*); a = 0.106 (0.000), b = 0.043 (0.000), c = 0.083 (0.000), d = 0.069 (0.000), e = 0.042 (0.000).

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
