# Peer review of "Association between Dental Treatment, Quality of Life, and Activity Limitation According to Masticatory Discomfort: Evidence from the Korean National Health and Nutrition Examination Survey (2013–2015)"

_ijerph, 2020, doi:10.3390/ijerph17020547_

Round 1
Reviewer 1 Report
RE: Association between Dental Treatment, Quality of Life, and Activity Limitation According to Masticatory Discomfort: Evidence from the Korean National Health and Nutrition Examination Survey (2013-2015) by Hyun-Kyung Kang and Yu-Rin Kim
The reviewer’s concerns are as follows
Major concerns
There are many studies investigating association of QOL and activity limitation with masticatory parameters. What is the novelty of this study? This is a cross-sectional study. However, several sentences not suitable for such study are seen throughout the whole manuscript. For example, in the Abstract, “Therefore, the 6th Korea National Health and Nutrition Examination Survey (KNHANES) data was used to identify the types of dental treatment and quality of life that influenced masticatory discomfort.” It is not appropriate to use “influence”. Also, in the Abstract, “The results showed that preventive measures and caries treatment can reduce the risk of masticatory discomfort and that recovery is difficult when oral disease progresses.” It is not obvious that preventive measures and caries treatment can actually reduce the risk based on the cross-sectional study.
Minor concerns
P2L3 from the bottom: How did the authors evaluate economic activity? The definition on masticatory discomfort should be clearly written. Table 3: What is “Old age”? The p-value can be written in a single line? How was the result of the other questions? The conclusion: “Since masticatory discomfort was identified to cause systemic activity limitation and decline in QoL, insurance benefit policies at the government level for primary prevention should be pursued actively to prevent masticatory discomfort from occurring.” It might not be “conclusion”.
Author Response
Reviewer 1
What is the novelty of this study? : Study on the masticatory discomfort is mostly directed at the elderly. However, our study expanded the subjects to all age groups and used national data representing Korea. Therefore, our study is very meaningful. However, several sentences not suitable for such study are seen throughout the whole manuscript. : We revised the abstract as the reviewer commented. Thank you. It is not obvious that preventive measures and caries treatment can actually reduce the risk based on the cross-sectional study. : We revised the abstract as the reviewer commented. Thank you. How did the authors evaluate economic activity? The definition on masticatory discomfort should be clearly written. : We have revised in details of economic activity and masticatory discomfort in the methods of the study. Table 3: What is “Old age”? The p-value can be written in a single line? How was the result of the other questions? The p-value of Table 4 is the value for the model suitability of the linear regression analysis. We presented many variables in a single table. So the p-values are attached at the bottom of the table : We deleted the "Old age" The conclusion: “Since masticatory discomfort was identified to cause systemic activity limitation and decline in QoL, insurance benefit policies at the government level for primary prevention should be pursued actively to prevent masticatory discomfort from occurring.” It might not be “conclusion”. : We revised the abstract as the reviewer commented. Thank you.Thank you very much for your opinion. We will strive for better study.
Thank you very much.

Reviewer 2 Report
This article is a good attempt at describing association between dental treatment, quality of life and activity limitation, but it is written in an awkard manner with several grammatical errors. Also, flow of the ideas is mixed up throughout the article and it can be improved. The results seem to be very obivious in that those who received dental treatment had less masticatory discomfort compared to those who did not. I am unsure whether this provides any new finding to the research world. It will be good to focus on unique findings if there were any from this research survey.
Abstract: The last sentence talks about active government salary policy for first prevention treatment, but this is not described anywhere in the article. I am not sure what active government salary policy means.
Introduction: Abstract talks about impact of masticatory discomfort on general health, but the introduction does not describe the impact adequately. Also, introduction mentions that masticatory ability can be assessed objectively and subjectively but this is not described and elaborated anywhere else in the article. The authors mention that masticatory discomfort causes significant deterioration of digestive health but fail to specify how the digestive function is deteriorated due to masticatory discomfort?
Methods: Types of dental treatment is not clear in the methods. It just mentions that 7 different treatments were investigated, but fails to specify and describe the treatments and also why these treatments are significant compared to others in the association of masticatory discomfort.
Results: Methods and results mentions economic activities, what are these economic activities and how do they impact? Many of the statistical results shows statistical significance but wondering if there is a large sample bias here. It would have been beneficial to calculate effect size to determine true impact of dental treatments, activity limitations and quality of life. Tables refer 'comfort' and 'discomfort'. It would be better to use 'No discomfort' and 'discomfort'.
Discussion: MT index and DT index were not mentioned in the article anywhere and it suddenly appears in the discussion. Unable to comprehend the sentence "According to the NHNES by the Korea Institute for Health and Social Welfare, almost half (46.6%) of the elderly population reported masticatory discomfort [19], but in the present study, there were more people without masticatory discomfort than those with masticatory discomfort, regardless of age. It is believed that such results reflect improved oral health status among the elderly population from expanded national health insurance coverage". How is the age related to improved oral health? Again, disucssion mentions economic activities without describing what they are and how they impact masticatory discomfort.
Conclusion: Conclusion mentions "since masticatory discomfort was identified to cause systemic activity limitation and decline in QoL, insurance benefit policies at the government level for primary prevention should be pursued actively to prevent masticatory discomfort from occurring" but how does insurance benefit at government level for primary prevention improve this. Is there any data that supports it other than your study. There is no description of this fact in the results or discussion.
Best wishes to the authors.
Author Response
Reviewer 2
It is written in an awkward manner with several grammatical errors.: We asked an expert to edit English grammar.
I am unsure whether this provides any new finding to the research world. It will be good to focus on unique findings if there were any from this research survey.
: Study on the masticatory discomfort is mostly directed at the elderly. However, our study expanded the subjects to all age groups and used national data representing Korea. Therefore, our study is very meaningful.
Abstract: The last sentence talks about active government salary policy for first prevention treatment, but this is not described anywhere in the article. I am not sure what active government salary policy means.
: We revised the abstract as the reviewer commented. Thank you.
Introduction: Abstract talks about impact of masticatory discomfort on general health, but the introduction does not describe the impact adequately.
: We revised the abstract as the reviewer commented. Thank you.
Also, introduction mentions that masticatory ability can be assessed objectively and subjectively but this is not described and elaborated anywhere else in the article.
: We revised the abstract as the reviewer commented. Thank you.
The authors mention that masticatory discomfort causes significant deterioration of digestive health but fail to specify how the digestive function is deteriorated due to masticatory discomfort?
: We deleted it because we thought it was unnecessary.
Methods: Types of dental treatment is not clear in the methods.
: We revised the abstract as the reviewer commented. Thank you.
It just mentions that 7 different treatments were investigated, but fails to specify and describe the treatments and also why these treatments are significant compared to others in the association of masticatory discomfort.
: We revised the abstract as the reviewer commented. Thank you.
Results: Methods and results mentions economic activities, what are these economic activities and how do they impact?
: We revised the abstract as the reviewer commented. Thank you.
Many of the statistical results shows statistical significance but wondering if there is a large sample bias here
: In order to reduce sample bias, we created a collective variable and analyzed it as a parent group when selecting a case. In addition, group variables were included in hierarchy variables when the analysis plan file was created. The sample bias was not large and We will consider your opinion in future research and try better study. Thank you.
It would have been beneficial to calculate effect size to determine true impact of dental treatments, activity limitations and quality of life.
: We will consider your opinion in future research and try better study. Thank you.
Tables refer 'comfort' and 'discomfort'. It would be better to use 'No discomfort' and 'discomfort'.
: We revised the 'No discomfort'
Discussion: MT index and DT index were not mentioned in the article anywhere and it suddenly appears in the discussion
: We deleted the MT index and DT index.
How is the age related to improved oral health? Again, discussion mentions economic activities without describing what they are and how they impact masticatory discomfort.
: We revised the abstract as the reviewer commented. Thank you.
Conclusion: Conclusion mentions "since masticatory discomfort was identified to cause systemic activity limitation and decline in QoL, insurance benefit policies at the government level for primary prevention should be pursued actively to prevent masticatory discomfort from occurring" but how does insurance benefit at government level for primary prevention improve this.
: We revised the abstract as the reviewer commented. Thank you.
Thank you very much for your opinion. We will strive for better study.
Thank you very much.

Round 2
Reviewer 1 Report
The manuscript was revised so well.
The minor concern is as follows;
This is the cross-sectional study, in which the cause-effect relationship can't be ascertained. So, the sentences in the conclusion, "Based on the findings of the present study, it was concluded that masticatory discomfort was reduced by preventive and curative caries treatment" and "Moreover, masticatory discomfort was found to be causing limitations in social activities and decline in QoL" might not be suitable.
Author Response
Reviewer 1
The manuscript was revised so well.
The minor concern is as follows;
This is the cross-sectional study, in which the cause-effect relationship can't be ascertained.
So, the sentences in the conclusion, "Based on the findings of the present study, it was concluded that masticatory discomfort was reduced by preventive and curative caries treatment" and "Moreover, masticatory discomfort was found to be causing limitations in social activities and decline in QoL" might not be suitable.
Answer
We agree with your opinion because our study is a cross-sectional study and an analysis of the results for that year. So, we modified it.
“Based on the findings of the present study, the discomfort of masticatory tended to be reduced due to preventive and early dental treatment and management.
In addition, the term ‘found’ changed to ‘showed’.
We modified it with a slightly lighter expression.
Thank you very much for your consideration.
We are looking forward to hearing from you positively
Thank you

Reviewer 2 Report
I commend the authors for making an effort to make changes to their article based on the reviewers comments. I would highly recommend reading through the article again and making grammer changes and make the flow of the article better since it appears to be piecemealed.
I still don't see what an economic activity is and there is no explanation or definition given. Does an economic activity is someone who has a full-time job?
Although, I understand that different age groups were included in the study, the end result of those who received preventive oral care had better clinical outcomes is not novel.
Best wishes to the authors.
Author Response
Reviewer 2
I commend the authors for making an effort to make changes to their article based on the reviewers comments.
I would highly recommend reading through the article again and making grammer changes and make the flow of the article better since it appears to be piecemealed.
I still don't see what an economic activity is and there is no explanation or definition given.
Does an economic activity is someone who has a full-time job?
Although, I understand that different age groups were included in the study, the end result of those who received preventive oral care had better clinical outcomes is not novel.
Answer
The revised paper was compiled in English by a native speaker.
Nevertheless, if there is any shortage, we would like to request the English editing of your journal (International Journal of Environmental Research and Public Health).
We have redefined the definition of economic activity used in this study.
“Economic activity is any individual activity related to the production and consumption of goods or services, or the distribution of income or wealth. Therefore, the difference between doing economic activity and not doing it is simply the difference between having income.”
We fully acknowledge your opinion.
However, most of the research was done on senior citizens in South Korea, and none of them were for all age groups. Therefore, I think our study on the whole age group in Korea is very meaningful.
In accordance with your opinion, we supplemented the contents with the discussion.
I deeply appreciate your opinion.
We are looking forward to hearing from you positively.
Thank you
